# The Prevalence of Invasive Bacterial Infection in Febrile Infants Presenting to Hospital Following Meningococcal B Immunisation: A Case Series

**DOI:** 10.3390/pediatric17010020

**Published:** 2025-02-08

**Authors:** Holly Drummond, Etimbuk Umana, Clare Mills, Thomas Waterfield

**Affiliations:** 1Wellcome Wolfson Institute of Experimental Medicine, Queen’s University Belfast, Belfast BT9 6BL, UK; eumana01@qub.ac.uk (E.U.); clare.mills@qub.ac.uk (C.M.); t.waterfield@qub.ac.uk (T.W.); 2Emergency Department, Royal Belfast Hospital for Sick Children, Belfast BT12 6BA, UK

**Keywords:** fever, infant, immunisation, post-vaccination fever, bacterial infections, bacteraemia

## Abstract

Objectives: To report the prevalence of invasive bacterial infection (IBI) in febrile infants ≤90 days presenting to hospital within 72 h of meningococcal B (MenB) immunisation. Methods: A secondary analysis of data from two previous multicentre studies of febrile infants conducted at UK and Irish hospitals. The first study was a retrospective study, conducted at six sites between 31 August 2018 and 1 September 2019. The second study was a prospective study conducted at 35 sites between 6 July 2022 and 31 August 2023. Febrile infants ≤90 days who had received the MenB vaccine within 72 h preceding presentation were included. Results: A total of 102 infants met the inclusion criteria, with a median age of 61 days and a male predominance of 65.7%. The most reported clinical features were an abnormal cry, decreased feeding and coryzal symptoms. In total, 68/102 (66.7%) were admitted to hospital; the median length of stay was 1 day. Median C-reactive protein (CRP) was 20.5 mg/L, mean white cell count was 13.7 × 10^9^/L, mean neutrophil count was 7.3 × 10^9^/L and mean lymphocyte count was 4.7 × 10^9^/L. In total, 38/102 (37.3%) had blood cultures performed, 26/102 (25.5%) had respiratory viral testing performed, 55/102 (53.9%) had urine culture performed and 14/102 (13.7%) had lumbar puncture performed. Additionally, 26/102 (25.5%) received parenteral antibiotics. There were no cases of IBI, and 3/102 (2.9%) cases of urinary tract infection. Conclusions: The rate of IBI is negligible in febrile infants following MenB immunisations. Current blood tests such as CRP are unreliable in this cohort, as many exhibit a moderate CRP rise above suggested international cut-offs for this age range.

## 1. Introduction

From September 2015, the meningococcal B (MenB or Bexsero) immunisation was introduced into the routine United Kingdom (UK) immunisation schedule. At two, four and 12 months of age, infants are offered the MenB vaccine, which has been reported to protect against approximately 88% of MenB strains [1]. Since the introduction of this vaccine, there has been an increase in infants presenting to hospital with post-vaccination fever [2]. A fever >38 °C has been reported in 51–61% of infants following administration of the MenB vaccine during routine immunisations [3]. Parents and guardians are advised to administer prophylactic paracetamol at the time of vaccination, followed by two additional doses at four to six hourly intervals [4]. Furthermore, parents and guardians are advised to seek medical care if the infant continues to appear unwell following vaccination [4].

The management of infants aged 90 days and younger presenting to hospital with fever can be challenging. These febrile infants are at a higher risk of invasive bacterial infection (IBI) compared to older children, with several studies reporting the rate of IBI at 1–4% in this cohort [5,6,7]. There are several clinical guidelines that provide varying recommendations for the management of febrile infants (Table 1). In the UK, there are two guidelines from the National Institute for Health and Care Excellence (NICE) that are widely used for the management of febrile infants. Namely, these are ‘Fever in under 5 s: assessment and initial management’ (NICE NG143) and ‘Sepsis: recognition, diagnosis and early management’ (NICE NG51) [8,9]. NG143 recommends that all febrile infants should have blood samples taken for full blood count, C-reactive protein (CRP) and blood culture [8]. Additionally, this guideline recommends that a lumbar puncture be performed without delay and parenteral antibiotics administered if the infant appears unwell [8]. NG51 recommends that all febrile infants be admitted to hospital and treated with parenteral antibiotics without delay, regardless of age, clinical features or laboratory results [9]. International guidelines such as those from the American Academy of Pediatrics (AAP) and the European Step-By-Step approach support a sequential assessment [6,7,10]. These guidelines advise that well-appearing infants aged 28 days and over undergo tailored investigations and treatments based on biomarker testing. However, they recommend that unwell-appearing and younger infants should be treated with parenteral antibiotics without delay [6,7,10].

The current guidelines do not make any recommendations for the management of febrile infants presenting following MenB immunisations [6,7,8,9,10]. Despite this, there is evidence that infants presenting with fever post-immunisation are at a lower risk of IBI [5,11,12,13]. It is likely that infants presenting following MenB immunisation may not require the extensive workup that febrile infants typically receive [13]. The aim of the present study is to report the prevalence of IBI in febrile infants aged 90 days and younger presenting to hospital within 72 h of receiving MenB immunisations.

## 2. Materials and Methods

### 2.1. Study Design

This study is a secondary analysis of data from two previous UK multicentre studies of febrile infants. The first study, a retrospective observational cohort study, has been previously described and is registered at ClinicalTrials.gov (NCT04196192) [5]. The study was conducted at six UK and Irish tertiary paediatric Emergency Departments (ED) selected from the Paediatric Emergency Research in the UK and Ireland (PERUKI) network, with one participating centre in Northern Ireland, one in the Republic of Ireland, one in Scotland and three in England. The study was conducted between 31 August 2018 and 1 September 2019 [5].

The second study was a prospective observational cohort study, the Febrile Infant Diagnostic Assessment and Outcome (FIDO) study. The study has been described previously and is registered at ClinicalTrials.gov (NCT05259683) [14,15]. The FIDO study was conducted at 35 PERUKI sites, with 30participating sites in England, two in Scotland, one in Wales, one in the Republic of Ireland and one in Northern Ireland. The study was conducted between 6 July 2022 and 31 August 2023 [14,15].

### 2.2. Study Population

The inclusion criteria for both previous studies encompassed infants aged 90 days and younger with fever (≥38 °C) attending UK or Irish hospitals. Infants whose guardians declined or withdrew consent were excluded. Infants recruited from either study who had received MenB immunisations within the preceding 72 h of presentation were included in this secondary analysis. Infants who had not received MenB immunisations within the preceding 72 h of presentation were excluded.

### 2.3. Outcome Measures

The primary outcome measure was to report the rate of IBI in infants presenting with fever within 72 h of receiving MenB immunisations.

Secondary outcome measures were to report the following:Urinary tract infection (UTI);Associated clinical features;Hospital admission;Length of stay;Investigations and procedures performed;Median CRP value;Mean white cell count (WCC);Mean absolute neutrophil count;Mean lymphocyte count;Parenteral antibiotic use.

### 2.4. Reference Standards and Definitions

Definitions were based on published standards [6,10,15]. IBI was defined as bacterial meningitis or bacteraemia (non-contaminant) confirmed by positive culture or quantitative polymerase chain reaction for a bacterial pathogen from blood or cerebrospinal fluid. The detection of coagulase negative *Staphylococcus*, *Streptococcus viridans*, *Propionibacterium acnes*, or Diphtheroids were considered contaminants. Hospital records were checked to identify unplanned return attendances, and these infants were assumed not to have IBI provided they did not receive an IBI diagnosis within seven days of discharge.

UTI was established by any of the following:Pure growth >100,000 CFU/mL of a single organism from a single clean catch urine sample.Pure growth >10,000 CFU/mL of a single organism from either a trans-urethral bladder catheter or suprapubic aspiration urine sample.Pure growth, from two urine samples, >100,000 CFU/mL of the same organism from urine pad/bag samples and the presence of pyuria (>10 white blood cells per high-power field) on laboratory microscopy.

Change in behaviour was defined as a noticeable alteration in usual actions such as increased irritability, excessive crying, or lethargy. Decreased feeding was characterized by less frequent feeding, shorter feeding duration, or refusal to eat. Decreased urine output was characterized by a reduction in wet nappies. Coryzal symptoms were defined as the presence of a runny nose, sneezing, and/or moist mucus membranes. Abnormal response to social cues was defined as an unusual reaction to typical social signs including delayed responses or unusual gaze patterns. Abnormal cry was defined as a cry that was unusual in pitch, pattern or duration, compared to a typical cry. An unwell appearance was defined as an abnormal global assessment or abnormal vital signs at presentation.

### 2.5. Study Procedures

For the retrospective study, patients were identified by searching emergency clinical software databases for all infants aged 90 days and younger presenting with a febrile illness [5]. The study included anonymised, non-personal, routinely collected clinical data only.

For the prospective FIDO study, patients were screened by trained clinical and research staff using a case report form (CRF) [14,15]. CRF data were recorded contemporaneously by the clinical staff or research teams prior to consent discussions. Research without prior consent methodology were used due to the emergency nature of IBI. This methodology is described in full in the FIDO study protocol [14,15]. In all instances, routine care was not interrupted or delayed.

### 2.6. Data Management

In the retrospective and prospective studies, data were collected and managed using REDCap (Research Electronic Data Capture) tools [16]. In the retrospective study, participants with incomplete clinical assessment data were excluded from the analysis [5]. In the prospective FIDO study, the initial CRF was completed during emergency care by the clinical staff or local research teams to record data including clinical presentation and initial examination findings [14,15]. The second CRF was completed seven days after discharge and recorded data including investigation results, length of stay, treatments given and unplanned return attendances. The CRF were uploaded to the REDCap on the Queen’s University Belfast servers.

### 2.7. Research Ethics

Based on the results of the Health Research Association decisions tool, national research ethics committee approval was not necessary for the retrospective study [5,17]. However, this study was registered and approved by the research governance offices at the respective sites. The FIDO study was given approval by the following research committees: Office for Research Ethics Committees Northern Ireland (ORECNI)—Health and Social Care Research Ethics Committee (HSC REC B, reference 22/NI/0002), Public Benefit and Privacy Panel for Health and Social Care (HSC-PBPP) Scotland (reference 2122-0257), and Children’s Health Ireland Research and Ethics Committee, Ireland (reference: REC-082-22) [14,15].

### 2.8. Statistical Analysis

Descriptive statistics were used to report the demographic characteristics, clinical features, IBI rates, lengths of stay and laboratory results. For categorical variables, frequencies and percentages were used. For continuous variables, mean and standard deviation (SD) were used for normally distributed data (based on the Shapiro–Wilk normality test) while median and interquartile ranges (IQR) were used for non-normally distributed data. Analysis was performed using GraphPad Prism^®^ version 10.0.0 for Windows (GraphPad Software, Boston, MA, USA).

## 3. Results

A total of 4025 infants were screened, of which 1590 were ineligible, 59 had incomplete data sets and 2376 were included (Figure 1). A total of 102 infants met the study inclusion criteria and had presented to hospital within 72 h following MenB immunisation. The median age of the study population was 61 days (IQR 58–69; range 44–89), with a male predominance of 65.7% (Table 2). Of the 102 infants, 84 (82.4%) had a fever ≥38 °C upon presentation to hospital. The most reported associated clinical features were an abnormal cry, decreased feeding and coryzal symptoms.

Of the 102 infants, 68 (66.7%) were admitted to a hospital ward (Figure 1). The median length of stay for those admitted was 1 day (IQR 0–2; range 0–4). Of the 34 (33.3%) patients who were not admitted to a ward, there was one unplanned return attendance within seven days of discharge. This infant did not receive a subsequent diagnosis of UTI or IBI. Of the 102 infants, 26 (25.5%) received parenteral antibiotics, either at the ED/assessment unit or once admitted to a ward.

Of the 102 infants, 44 (43.1%) patients had blood tests performed, and the CRP values were reported (Table 2). The median CRP (n = 44) was 20.5 mg/L (IQR 5.0–38.5). The mean ± SD WCC (n = 42) was 13.7 × 10^9^/L ± 4.5 × 10^9^/L, the mean ± SD neutrophil count (n = 42) was 7.3 × 10^9^/L ± 2.8 × 10^9^/L, and the mean ± SD lymphocyte count (n = 42) was 4.7 × 10^9^/L ± 1.9 × 10^9^/L. Of the 102 patients, 38 (37.3%) had blood cultures performed. There were no cases of bacteraemia. There were 2 blood cultures with confirmed contaminants (1 *Staphylococcus epidermidis* and 1 coagulase negative *Staphylococcus*). In total, 26 (25.5%) patients had respiratory viral testing performed, of which 9 were positive for at least 1 viral pathogen. Additionally, 55 (53.9%) patients had urine culture performed, and 3 were diagnosed with UTI (2 *Escherichia coli* and 1 *Klebsiella*). A total of 14 (13.7%) infants had lumbar puncture and cerebrospinal fluid (CSF) culture performed, all of which were negative for bacterial pathogens. For 7 of the 14 infants receiving lumbar puncture, the number of attempts were recorded, which ranged from 1 to 9 attempts. The median number of lumbar puncture attempts were 4 (IQR 3–4). Overall, there were no cases of IBI and 3 cases of UTI within the cohort.

## 4. Discussion

This secondary analysis of 102 febrile infants, aged 90 days and younger, presenting to hospitalwithin 72 h of MenB immunisations found no cases of IBI. Several other studies have reported similar findings, including two prospective studies of febrile infants presenting to hospital following MenB immunisations [11,12,13,18,19]. Despite this consistent finding, national guidelines in the UK, Europe and the United States of America do not provide advice for the management of infants presenting to hospital within 72 h of MenB immunisations.

Without specific guidance, the management of these infants is challenging. This was demonstrated by 43.1% of infants undergoing blood tests, 66.7% of infants being admitted, 13.7% undergoing lumbar puncture and 25.5% receiving parenteral antibiotics, despite no child being diagnosed with IBI. These results reflect the findings of other, similar studies reporting admission rates of 51–72% and lumbar puncture rates of 16–17% [18,19]. It is this typically cautious approach that leads to the high healthcare costs. It has been reported that, in the UK, ED resource use costs are £1000.28 (95% CI £82.39–£2993.37) per infant, much higher than any other paediatric age group [20].

Many clinical guidelines, such as AAP and Step-By-Step, now include specific criteria for identifying lower-risk cohorts of infants that may not require lumbar puncture and parenteral antibiotics [6,7,10]. These guidelines all recommend a CRP cut-off of 20 mg/L, above which all infants should be deemed ‘high risk’. In this analysis, however, the median CRP was 20.5 mg/L, above the cut-off and likely reflecting a normal immune response to immunisation. It is likely that CRP at the current cut-off recommendations is not optimised for the risk-assessment of febrile infants presenting within 72 h of MenB immunisations.

UTIs are the most common serious bacterial infection (SBI) in febrile infants aged 90 days and younger, accounting for approximately 80% of SBIs [5,6,10,21]. In the present study, there were 3 (2.9%) infants with UTI, which is lower than the 10–20% typically reported in cohort studies of febrile infants [5,6,10,21]. This non-negligible risk of UTI indicates that urinalysis testing may still be useful in this cohort.

Finally, despite the existing national guidelines, there does appear to be a shift in practice, with clinicians already choosing to safely do less. In this secondary analysis, 13.7% of infants received a lumbar puncture and 25.5% received parenteral antibiotics. These are lower numbers than the proportions reported from the parent studies, with 59% and 52% undergoing lumbar puncture in the retrospective and prospective cohorts, respectively, and 76% and 68% receiving parenteral antibiotics in the retrospective and prospective cohorts, respectively [5,15].

Although this is one of the largest cohorts of infants presenting to hospital post-immunisation, the total number of participants remains small. Ultimately, a systematic review and meta-analysis is required in order to provide a definitive answer. Additionally, this secondary analysis included data from a retrospective study, which limited the available data to the information in the medical chart. Furthermore, the exact timing of the immunisations was not recorded and exact timing of presentation at hospital post-vaccine could not be reported.

## 5. Conclusions

In this secondary analysis, the rate of IBI in febrile infants presenting to hospital following MenB immunisation was 0%. Infants presenting following routine MenB immunisations may not require the extensive investigations that febrile infants typically receive. Safely reducing the number of procedures performed is likely to result in less distress for the child, improve antimicrobial stewardship, and has potential cost savings. Current clinical guidelines do not take infants with post-vaccination fever into account. Further research with larger study populations is necessary to support the development of guidelines.

## Figures and Tables

**Figure 1 pediatrrep-17-00020-f001:**
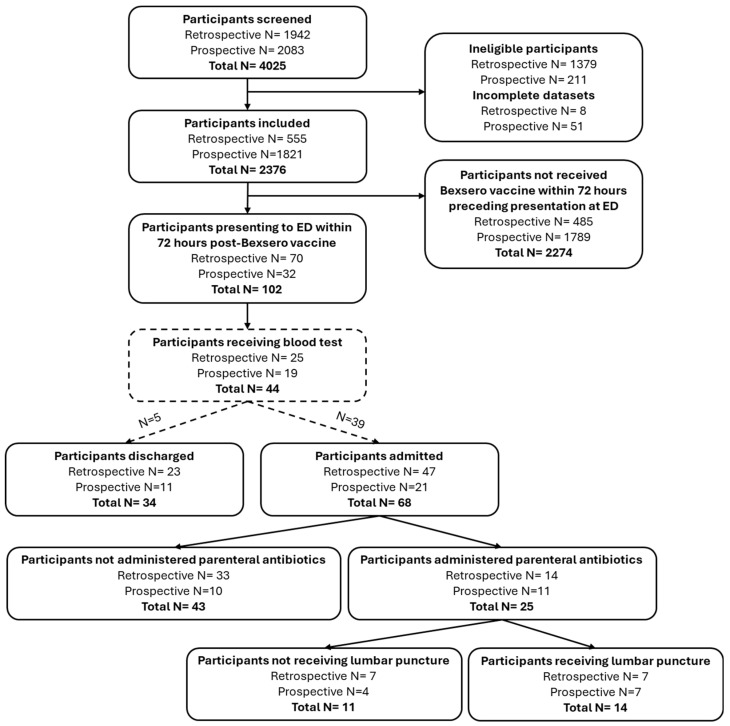
Flow of participants through the study.

**Table 1 pediatrrep-17-00020-t001:** Variations in clinical guidelines for the management of febrile infants.

Guideline	Age Range	Biomarkers	Exclusions
NICE	≤90 days	CRP, WCC, ANC	None
AAP	≤60 days	PCT, CRP, ANC	Preterm
			<2 weeks old, perinatal course complicated
			Suspicion of HSV infection
			Focal bacterial infection
			Clinical bronchiolitis
			Immuno-compromised
			Neonatal course complicated
			Congenital/chromosomal abnormalities
			Requires technology/therapeutic intervention to sustain life
			Immunisations within preceding 48 h
Step-By-Step	≤90 days	PCT, CRP, ANC	Clear source of fever Preterm
			Unexplained hyperbilirubinemia
			Hospitalised longer than mother
			Receiving current/previous antimicrobial therapy
			Previous hospitalization
			Chronic/underlying illness

NICE, National Institute for Health and Care Excellence; AAP, American Academy of Pediatrics; CRP, C-reactive protein; WCC, white cell count; ANC, absolute neutrophil count; PCT, procalcitonin; HSV, herpes simplex virus.

**Table 2 pediatrrep-17-00020-t002:** Characteristics of patient population and investigations performed.

Infant Characteristic	Results (n = 102), N (%)
Age, days	
Median (IQR; range)	61 (58–69; 44–89)
Gender	
Male	67 (65.7)
Female	35 (34.3)
Temperature, °C	
39–39.9	8 (7.8)
38–38.9	76 (74.5)
37–37.9	11 (10.8)
<37	7 (6.9)
Associated clinical features *	
Change in behaviour (irritability, crying, lethargy)	18 (17.6)
Decreased feeding	34 (33.3)
Decreased urine output	5 (4.9)
Coryzal symptoms	30 (29.4)
Abnormal response to social cues	4 (3.9)
Abnormal cry	53 (52.0)
Swollen joint/limb present	1 (1.0)
Unwell-appearing	22 (21.6)
Investigations	
C-reactive protein, mg/L (n = 44)	
Median (interquartile range; range)	20.5 (5.0–38.5; 0.7–83.0)
White cell count (×10^9^/L) (n = 42)	
Mean ± standard deviation (range)	13.7 ± 4.5 (3.7–23.7)
Neutrophil count (×10^9^/L) (n = 42)	
Mean ± standard deviation (range)	7.3 ± 2.8 (1.9–14.9)
Lymphocyte count (×10^9^/L) (n = 42)	
Mean ± standard deviation (range)	4.7 ± 1.9 (1.0–9.2)
Blood culture (n = 38)	
Positive	2 (contaminants)
Negative	36
Respiratory viral testing (n = 26)	
Positive	9
Negative	17
Urine culture (n = 55)	
Positive	3
Negative	52
Cerebrospinal fluid culture (n = 14)	
Positive	0
Negative	14

* Can have more than one associated clinical feature.

## Data Availability

All the data used in this study are available, as detailed in the original study publications [5,14,15].

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
