# Peer review of "The Prevalence of Invasive Bacterial Infection in Febrile Infants Presenting to Hospital Following Meningococcal B Immunisation: A Case Series"

_pediatrrep, 2025, doi:10.3390/pediatric17010020_

Round 1
Reviewer 1 Report
Comments and Suggestions for Authors
The manuscript titled "The prevalence of invasive bacterial infection in febrile infants presenting to hospital following meningococcal B immunisation: a case series" presents a secondary analysis of data from two previous multicenter studies of febrile infants ≤90 days of age presenting to the hospital within 72 hours of meningococcal B (MenB) immunization, focused on reporting the prevalence of invasive bacterial infection (IBI). The authors found a 0% prevalence of IBI in these patients, arguing persuasively that the generalized application of NICE guidelines may lead to over-diagnosis, over-treatment, and unnecessary healthcare costs due to the lack of tailored guidance for this subpopulation, confirming previous findings. Guidelines from other countries were cited as a possible example of a better, more tailored approach for this type of subpopulations that may not require excessive and invasive testing. The article is well organized and clearly written; the authors used robust methods and the statistical approach outlined in the study is appropriate for the stated aims. The authors also adequately address the study’s inherent limitations, including the small cohort size, the secondary nature of the analysis, and the lack of precise timing data for vaccination and hospital presentation. Overall, this study addresses a timely and clinically significant question. These findings are highly relevant to pediatricians and emergency care providers.
Reviewer 2 Report
Comments and Suggestions for Authors
Dear Editor,
I thank you for the opportunity you have given me to review this manuscript written by Drummond et al.
I read the study with great interest, as it concerns a topic of certain scientific interest that has unjustifiably not yet been adequately addressed in the literature dedicated to the management of febrile infants.
I am not a native English speaker, so I do not give suggestions on vocabulary and grammar. I was able to read the text with ease and no section of the article is unclear.
Here are my observations:
- line 40: the Authors have correctly reported the definition of IBI, but only later in the article, SBI and UTI are defined. It could be useful to report the definitions in the introduction to make the article more readable even for those who are not fully familiar with the literature on the topic discussed.
- line 50-55: the Authors correctly report the main protocols dedicated to the management of febrile infants, in addition to the NICE guidelines. However, I think they should specify, at least in a short sentence, that these three protocols differ greatly from each other, both in terms of inclusion criteria (for example the cut off < 60 or < 90 days of life) and in terms of parameters taken into consideration (one above all, procalcitonin which is not considered in the NICE guidelines). Finally, they should report that fever arising within 48 hours of vaccination is an exclusion criterion for AAP.
lines 151 and on: I noticed a non-negligible percentage of hospitalizations (66.7%) of patients admitted for vaccine fever. Is it possible that this depends on the age at admission and that some patients were subjected to the first dose of the vaccine before completing the 60th day of life, further worrying the physician?
line 173: I note an incredibly high percentage of patients subjected to lumbar puncture. This may depend on the substantial differences present, for example between the Step-by-Step approach, which we follow, and the NICE guidelines. Trying to apply the NICE guidelines a posteriori, and imagining that there were no patients < 30 days, the lumbar puncture would be justified by unwell clinical conditions or by leukocytosis/leukopenia. Is it possible, retrospectively, to give a justification for performing a lumbar puncture?
line 193: in addition to the high percentage of patients undergoing lumbar puncture, I also find the percentage of patients undergoing parenteral antibiotic therapy incredibly high. Is it perhaps due to a mean CRP value > 2 mg/l, which is univocally considered as a cut-off to discriminate the risk of an SBI/IBI?
line 203: the Authors correctly attribute the CRP value to the immunization expected after a vaccine. It would therefore be interesting to know the values of procalcitonin, which is now recognized as the most reliable marker for bacterial infection, but I believe that, since it is not foreseen by the NICE guidelines, the Authors do not have this data. The Authors should add a comment on this anyway.
line 205: a time window of 72h is incredibly wide, especially in febrile infants. It has been shown that tests performed too early can be falsely negative, and therefore could mislead the physician (doi: 10.1111/apa.16682). I suggest to the Authors, if they have not done so, to divide the enrolled patients based on the hours elapsed between the vaccine and the onset of fever, at the time of access to the ER. This data could further discriminate which patients may actually deserve further investigation and would help to better interpret the blood tests. If this data were not available, it could be cited in the limits suggesting a future analysis.
line 220: the Authors correctly state that despite the large case study, 102 cases could be too few to reach reliable conclusions. Can the Authors suggest a minimum number of patients to enroll in order to confirm 0% IBI in patients with vaccine fever?
Finally, a curiosity. Who perform the MenB vaccine in the UK and Ireland? Are there local health authorities with staff dedicated to administering the vaccine or are they administered by your GP? This is very useful because access to the ER of post-vaccination feverish infants can also depend largely on the uniformity of the information that is released to parents.
DOI: 10.1111/apa.16682
Reviewer 3 Report
Comments and Suggestions for Authors I would like to thank the authors for their submission and for providing me the opportunity to review their work. The topic of this study is both highly relevant and original within the field of pediatric infectious diseases and immunization. It specifically addresses an important gap in our understanding of the relationship between meningococcal B (MenB) immunization and the incidence of invasive bacterial infections (IBI) in febrile infants under 90 days old. While the safety and efficacy of vaccines are widely studied, the post-immunization period has not been thoroughly explored in the context of IBI in this specific age group. Given the critical nature of diagnosing and managing IBIs in such young infants, the findings of this study have the potential to directly inform clinical decision-making. By improving early detection practices and refining post-vaccination monitoring protocols, this research could help reduce unnecessary testing, thereby minimizing distress for the child and improving antimicrobial stewardship. The study’s conclusions directly address the central question of whether there is an increased risk of IBI in infants within 72 hours following MenB vaccination. The results show that the rate of IBI in this population is not significantly higher than in the general febrile infant population. As such, the conclusions align with the presented evidence, offering reassurance to clinicians about the safety of MenB immunization. Furthermore, the study suggests that febrile infants presenting after MenB vaccination may not require the extensive investigations that are typically performed in febrile infants. The references cited are appropriate, as they are both relevant to the topic and provide necessary context for the findings. However, I would appreciate it if the authors could address the following points to further strengthen the manuscript: 1) ABSTRACT: Page 1, line 15 I suggest specifying the mean age (± standard deviation), and sex of the study population. 2) INTRODUCTION: Page 1, line 32 I suggest offering a more comprehensive context about the significance of meningococcal B immunization for public health and the value of its extensive distribution. 3) INTRODUCTION: Page 2, line 54 I suggest providing a clearer definition of “unwell appearing” to reduce ambiguity. 4) MATERIALS AND METHODS: Page 2, line 62 I suggest clearly listing the exclusion criteria for the study. 5) RESULTS: Table 1, Page 5, line 181 I recommend providing a detailed definition of the various associated clinical features in the Materials and Methods section. 6) DISCUSSION: Page 7, line 225 I suggest providing a more detailed description of the future prospects and clinical implications of this study.
